# Study on Quasi-Static Axial Compression Performance and Energy Absorption of Aluminum Foam-Filled Steel Tubes

**DOI:** 10.3390/ma16124485

**Published:** 2023-06-20

**Authors:** Zhanguang Wang, Jianhua Shao

**Affiliations:** 1School of Architectural Engineering, Kaili University, Kaili 556011, China; wzg3262396@163.com; 2School of Civil Engineering and Architecture, JiangSu University of Science and Technology, Zhenjiang 212003, China

**Keywords:** aluminum foam-filled steel tube, numerical simulation, axial compression, carrying capacity, energy absorption

## Abstract

To study the axial compression performance of aluminum foam-filled steel tube and empty steel tube as objects, such tubes are studied in this paper, which explores the carrying capacity and deformation behavior of aluminum foam-filled steel tube with different lengths under a quasi-static axial load through experimental research. The carrying capacity, deformation behavior, stress distribution, and energy absorption characteristics of empty steel tubes and foam-filled steel tubes are compared through finite element numerical simulation. The results indicate that, compared with the empty steel tube, the aluminum foam-filled steel tube still presents a large residual carrying capacity after the axial force exceeds the ultimate load, and the whole compression process reflects steady-state compression. In addition, the axial and lateral deformation amplitudes of the foam-filled steel tube decrease significantly during the whole compression process. After filling the foam metal, the large stress area decreases and the energy absorption capacity improves.

## 1. Introduction

At present, there is a lot of research being carried out on energy-absorbing materials, components, and structural forms, including energy-absorbing analyses of building structure forms and components under wind load, earthquake load, and other loads [1,2,3,4], as well as energy-absorbing analyses of thin-walled steel tubes commonly used in the automobile and ship industries under impact. Thin-walled steel tube refers to a structure whose thickness is much less than its length and width. Because of its good energy absorption capacity, it has been widely used in the road transportation, rail transportation, aerospace, and ship industries [5,6,7]. Thin-walled steel tubes are also prone to deformation instability and an excessive initial peak during axial deformation due to their thin wall thickness [8]. When subjected to oblique loads, thin-walled structures are prone to overall buckling, which leads to reduced energy absorption [9]. To solve this kind of problem, researchers proceed from the structure of thin-wall steel tubes and improve the energy-absorbing capacity and stability of the structure by making holes [10], grooving [11] or pre-buckling [12], local heat treatment [13], and other treatment methods.

In addition, foamed metal materials with high energy absorption capacity during compressive deformation are becoming increasingly attractive in impact applications [14,15]. The foam metal is combined with the thin-walled steel tube to form a composite material. On the one hand, the internal deformation of the steel tube will be limited by the foam metal, so that the deformation can only occur within a certain range, greatly reducing the overall instability of the steel tube and improving its ability to absorb energy. On the other hand, due to the interaction between the foam material and the tube wall, the energy absorption capacity of the combined aluminum foam-filled steel tube is higher than the steel tube and aluminum foam under independent compression [16,17].

In general, the research on the compression performance of aluminum foam-filled steel tubes can be divided into quasi-static compression and impact compression. According to the different loading directions, it can be divided into the axial direction and the oblique direction. Zhang et al. [18], Liu et al. [19], and Li et al. [20] studied the energy absorption characteristics of thin-walled circular tubes with different sections under static axial load, as well as the influence of various factors on their compression behavior. Utilizing finite element analysis, Yang et al. [21] found that the energy absorption of multi-layer aluminum tubes, where a smaller aluminum tube is inserted inside the aluminum tube, was improved in comparison with single-layer aluminum tubes, and the specific energy absorption rate was also increased. Luo et al. [22], through experimental research, concluded that, on the premise of the same absorbed energy, foam-filled metal steel tubes could reduce the volume and mass required by energy-absorbing members compared with pure steel tubes. To explore the response of the foam-metal structure under the impact load, Dipen Kumar Rajak et al. [23] and Patel et al. [24] studied the crashworthiness and energy absorption performance of the foam-filled thin-walled steel tube under different strain rates. The results display that the strain rate increases linearly with the platform stress and energy absorption. Jia et al. [25] combined the smooth particle fluid dynamics program to carry out the numerical simulation of an ultra-high-speed impact and compared the crashworthiness and ultimate performance of various filled aluminum foam structures. It was found that the protection performance of the filled aluminum foam structure was better than that of the filled solid aluminum plate structure with the same surface density when the aluminum foam was in an appropriate position. Jin et al. [26] and Zhang et al. [18] proved through numerical simulation that the loading speed had a distinct influence on the total energy absorption of the foam-filled circular tube and that the total energy absorption increased with the increase in the loading speed. Generally, it is difficult for structures to control the load direction under impact loads, so research on the structural response under oblique loads are also carried out [27,28,29]. Huang et al. [28] utilized finite element software to systematically analyze the mechanical response, deformation regularity, and energy absorption characteristics of such parameters as the length, diameter, wall thickness, aluminum foam density, and impact velocity of thin-walled metal circular pipes and foam-filled pipes under the action of a lateral impact load. Wang et al. [30] studied the axial compression of aluminum foam-filled steel tube samples under different high temperatures. It was discovered that the axial compression failure mechanism and increasing the thickness of steel tubes can improve their bearing capacity. At the same time, compared with the rectangular section, the upper limit of the return platform of the square section bar is further increased.

Although aluminum foam-filled steel tubes have been studied and analyzed under axial load tests, there are few systematic analyses of the energy absorption capacity of components with different length specifications. Given this, this paper analyzes the compressive performance of aluminum foam-filled steel tube members with different length specifications through experimental research. The deformation characteristics, stress variation, and energy absorption capacity of aluminum foam-filled and empty steel tubes with different lengths and wall thicknesses are compared and analyzed through the finite element method.

## 2. Axial Compression Performance Test

In this chapter, in order to study the bearing capacity and deformation characteristics of foam-filled steel tube members under axial pressure, the quasi-static axial compression tests were carried out, and the force-displacement curves and deformation characteristic diagrams of steel tube members of different lengths under load were obtained. The influence of length parameters was obtained on the bearing capacity and deformation of foam steel tube.

### 2.1. Mechanical Properties of Galvanized Steel

The galvanized steel tube in this test exhibits excellent mechanical properties as well as corrosion resistance. The significant parameters of material mechanical properties are listed in Table 1, and the stress-strain curve is illustrated in Figure 1.

### 2.2. Performance Test of Foamed Metal Material

The foamed metal is made of aluminum foam and prepared by melt foaming. The porosity *P_r_* of cellular aluminum foam is:(1)$$P_{r}=1-m/(V\rho_{s})=1-\rho /\rho_{S}$$
where ρ is the apparent density of cellular foam metal, $\rho_{S}$ is the density of aluminum matrix material, set at 2700 kg/m^3^, and V is the volume of aluminum foam. Uniaxial compression tests were carried out on aluminum foam with different porosities to obtain the stress-strain curve of aluminum foam, as shown in Figure 2. Table 2 displays the material mechanical properties parameters of aluminum foams with three different porosity ratios.

### 2.3. Loading Scheme

The compression test was performed on an electronic universal testing machine (WDW2000) manufactured by Jinan Hengshang Instrument Co., Ltd. (Jinan, China), which is shown in Figure 3. The axial force-displacement curve with a displacement loading speed of 1 mm/min was collected by the data acquisition system.

### 2.4. Axial Compression Test of Aluminum Foam-Filled Steel Tube

The steel tubes tested were galvanized steel tubes with a rectangle section size of 40 mm × 40 mm, a wall thickness of 0.8 mm, and filled with aluminum foam with a porosity rate of 90%. Axial compression tests were carried out on the foam-filled steel tube members with a length of 120 mm, 200 mm, and 280 mm, respectively. The deformation of the specimen, which is obtained through the test, is shown in Figure 4.

It can be seen from the figure that the axial deformation of the three kinds of specimens is relatively regular, both of which display that the folds appear alternately in two directions of the section and that the folds appear first in the direction of the long dimension side, and that the deformation also concentrates on one end of the specimen. In combination with Figure 5, the axial force-displacement curve shows that the axial carrying capacity of the member reaches the peak load when the first fold appears, then decreases to a certain fixed value and fluctuates, and each load wave peak corresponds to a fold.

Figure 6 shows the comparison of peak loads obtained by the axial compression tests of members with different lengths. The peak loads of 120 mm, 200 mm, and 280 mm specimens are 41.05 kN, 31.2 kN, and 24.12 kN, respectively, which gradually decrease with the increase of axial compression specimen length; the average load of the specimens is 20.93 kN, 19.93 kN, and 17.58 kN, respectively, which also decreases with the increase of specimen length.

## 3. Numerical Simulation Model Validation

### 3.1. The Establishment of a Numerical Model

(1) Finite Element Model of Aluminum Foam

The foam metal uses a compressible foam material model [31], which can be used to simulate compressible foams that collide with other materials. The model is also related to the strain rate, and the Poisson’s ratio is assumed to be zero under unidirectional axial pressure. On the premise of using this algorithm, the elastic modulus is constant, and the stress is the elastic effect. The stress formula is as follows:(2)$$\sigma_{\mathrm{ij}}^{n+1}=\sigma_{\mathrm{ij}}^{n}+E{\overset{•}{\varepsilon }}_{\mathrm{ij}}^{n+1/2}\Delta t^{n+1/2}$$
where ${\overset{•}{\varepsilon }}_{\mathrm{ij}}$ is the strain rate; *E* is the elasticity modulus; and *t* is time.

In the compressible foam material model [32], there is a critical value of tensile stress used to define failure under tensile load. Below this critical value, the material is in an elastic, recoverable state under tensile and compression loads. The SOLID164 element in LS-DYNA is used to grid the foam metal material. SOLID164 is an explicit structural entity element for three dimensions, consisting of 8 nodes. The SOLID164 element can only be used for dynamic explicit analysis and supports all nonlinear properties.

(2) Finite Element Model of Steel Tube

Steel tube adopts the LS-DYNA piecewise linear Cowper–Symonds plastic model, which can input the stress-strain curve related to strain rate. The actual stress-strain curve of the material is input into the model, and the plastic criterion is applied to define the failure of the component according to the plastic strain.

Furthermore, the steel tube has been meshed with a shell 163 element, which is a 4 node explicit structural thin shell element. The element has the characteristics of bending and membrane, that is, it has certain stiffness in and out of the plane and can bear the load in the plane and the normal direction of the plane. 

### 3.2. Reliability Verification of the Numerical Simulation Method

The simulation data of the axial pressure of the foam-filled steel tube are compared with the experimental data of Wang [33] on the compression performance of the rectangular galvanized steel tube with foam fill. The axial pressure model of the foam-filled steel tube is shown in Figure 7. It is composed of (a) steel tube, (b) the top pressing part, (c) the bottom pressure-carrying component, and (d) aluminum foam. The specific material parameters are shown in Table 3.

The top pressing components are only allowed to move in the axial direction; their size is 50 mm × 50 mm × 10 mm. The consolidation constraint is applied to the bottom pressure-carrying component, whose dimensions are 50 mm × 50 mm × 10 mm. The section size of the steel tube is 40 mm × 40 mm, and the wall thickness and length are 1 mm and 120 mm, respectively. 

A small gap between the foamed metal and the steel tube is utilized to eliminate the negative volume and leave room for contact, and the final size of the aluminum foam was determined to be 39.5 mm × 39.5 mm × 120 mm. Self-contact is adopted for aluminum foam, and the foam metal and the top pressing component, the bottom pressure-carrying component, and the steel tube are all in the form of face-to-face contact. The coefficient of static friction and the coefficient of dynamic friction between contact surfaces are 0.3 and 0.2, respectively. To simulate the test state of quasi-static loading, a 50 mm displacement load was applied to the rigid body under the top, and the loading speed was kept at 1 mm/s. 

The results of the axial compression test and simulation data of foam-filled steel tubes are shown in Figure 8.

It can be learned from the figure that the change of the curve shows the same trend, and the maximum carrying capacity of the numerical simulation appears earlier, but it is roughly the same as the peak load value obtained from the test. The first peak load of the numerical simulation was 45.66 kN, and the peak load obtained by the test was 45.36 kN, with an error of 2.2%. The second peak point of the numerical simulation was 26.21 kN, and the second peak point obtained by the experiment was 25.10 kN, with an error of 4.4%. Compared with the axial force-displacement curve obtained by the test, the axial force-displacement curve obtained by the simulation was shifted forward. Because there was a certain gap between the top pressing component and the specimen in the test, and the load was very small under the premise of a displacement.

In combination with the deformation comparison diagram of numerical simulation and experiment in Figure 9, it can be deduced from the figure that the deformation presents approximately the same shape, which is in the form of staggered folding, with two folds appearing in one group of opposite edges and three folds appearing in the other group of opposite edges.

Through the comparison of the above numerical simulation and test data, it can be concluded that the axial force-displacement curve of the numerical simulation is relatively close to the test data, and the deformation of the two also presents a similar rule, that is, they all present the form of staggered and overlapped adjacent sides, which shows the accuracy of the numerical simulation method and lays a foundation for the subsequent parametric finite element analysis.

## 4. Comparative Analysis of Axial Compression Performance of Foam-Filled Steel Tube and Empty Steel Tube

In this chapter, the explicit dynamic analysis module of LS-DYNA in ANSYS finite element software is adopted to conduct a numerical simulation of aluminum foam-filled steel tube members under quasi-static axial compression. Specifically, the length and wall thickness of the foam-filled steel tube members are studied and compared with the empty steel tube. The axial force-displacement curve obtained by numerical simulation is given, and the effects of different length and wall thickness parameters are obtained on the bearing capacity and deformation of foam-filled steel tubes.

### 4.1. Comparative Analysis of Different Member Lengths

#### 4.1.1. Comparative Analysis of Bearing Capacity

To further enrich the research content of the test, the same section size as the test is taken, that is, the section of the steel tube is 40 mm × 40 mm, the wall thickness is 1 mm, and the length is 100 mm, 150 mm, 200 mm, and 250 mm, respectively, as the numerical simulation analysis object. The material parameters of the steel tube and aluminum foam are the same as those in Table 3. To simulate the test state of quasi-static loading, a 50 mm displacement load was applied to the top pressing component, and the loading speed was kept at 1 mm/s.

The axial force-displacement curves are shown for the empty steel tubes and the foam-filled steel tubes of different lengths in Figure 10. It can be observed that the carrying capacity of empty steel tube members decreases sharply after the first peak load, and the compression processes in the later period reflect unsteady compression. However, even though the axial force of the foam-filled steel tube exceeds the ultimate load, it still possesses a large residual carrying capacity. In the subsequent compression process, the two sides of the member are alternately folded, but the carrying capacity shows stable fluctuation and does not decrease all the time. The whole compression process involves steady-state compression.

Figure 11 depicts the peak load variation trend for various member lengths. In the figure, ET represents the peak load of an empty steel tube, FFT represents the peak load of the foam-filled aluminum tube, and 1, 2, 3, and 4, respectively, represent the peak loads corresponding to the first, second, third, and fourth peaks. As can be seen from the figure, the difference in the first peak load is not large for the two kinds of members, and both show a trend of decreasing with the increase in member length. Comparing the second peak load, it is found that the second peak load of pure steel tube is relatively small, whereas the second peak load after filling with aluminum foam has been increased compared with pure steel tube, and there is a tendency to increase with the growth of length. The third and fourth peak loads of the empty steel tube are close to each other, and neither of them has changed too much with the increased length of the member. However, compared with the empty steel tube, the third and fourth peak loads of the foam-filled steel tube are greatly increased, and the load tends to increase with member length, and the third and fourth peak loads are maximum when the member length is 200 mm. The first, second, third, and fourth peak loads of the empty steel tube decrease successively, while the first peak load of the foam-filled steel tube is the largest and the second peak load is the smallest. The third and fourth peak loads increase successively compared with the second peak load. This indicates that the foam-filled steel tube can significantly improve its buckling load capacity compared with an empty steel tube. 

Combined with Table 4, it can be concluded that the first peak load of the empty steel tube decreases gradually with the increase in member length. The influence of foamed metal was not significant on the first peak load, and the percentage of added value is 2.8%, 2.0%, 1.2%, and 3.9% for the first peak load of the foam-filled steel tube with different lengths, respectively, compared with the empty steel tube. By comparing the second peak load of the two members, it is discovered that the carrying capacity of the steel tube has been improved to a certain extent due to the aluminum foam-fill, and the members of different lengths have been increased respectively: 12.3%, 17.7%, 22.2%, and 51.1%. With the increase in axial loading displacement, the third peak load of the filled steel tube has a more obvious carrying capacity advantage. The carrying capacity of the filled steel tube components of different lengths has been increased: 51.2%, 51.2%, 86.1%, and 61.7%, respectively. The fourth peak load increased by 50.1%, 86.9%, 101.5%, and 85.4%, respectively.

#### 4.1.2. Comparative Analysis of the Deformation Pattern and Stress Cloud Diagram

The deformation pattern was compared and analyzed between a 200 mm length empty steel tube and an aluminum foam-filled steel tube. The deformation pattern is shown in Figure 12. As can be seen from Figure 12a, the deformation of the empty steel tube is shown by the form of adjacent staggered internal folding. Each fold is relatively flat in the cross-section, and the deformation amplitude of concave and convex around the cross-section is large, and there are obvious cavities in the deformation. As can be learned from Figure 12b, the deformation of the foam-filled steel pipe is similar to the adjacent staggered folding deformation. There were two folds on the opposite side of the two groups, and the deformation positions are at the end.

By comparing the deformation patterns of empty steel tubes and aluminum foam-filled steel tubes, it can be concluded that due to the lateral restriction of metal foam on steel tubes, the folded size and lateral deformation amplitude of the foam-filled steel tubes are smaller, and there is no cavity in the deformation of the steel tubes. In addition, the deformation regularity under axial compression becomes more stable after filling with foam metal, and the axial deformation interval also becomes smaller. 

The stress cloud diagram is used to compare and analyze the members of the 200 mm empty steel tube and the aluminum foam-filled steel tube. The stress cloud diagram is shown in Figure 13. As can be learned from the figure, the stress is also high in the part with the greatest deformation, and the maximum stress occurs at the corner of the section. This indicates that the stress concentration phenomenon will appear at the four right angles of the rectangular section member, which is also more prone to fail. The foam-filled metal presents a beneficial protective effect on the steel tube. In the figure, the large stress area of the foam-filled steel tube appears smaller than that of the empty steel tube. The stress of the foamed metal is also concentrated at the end of deformation, and the maximum stress also occurs at the corner.

#### 4.1.3. Comparative Analysis of Different Member Lengths

In the process of axial deformation, the absorption of energy is an important index to evaluate the mechanical properties of components. To better illustrate the energy-absorbing performance of members, some quantifiable indicators are introduced, as follows:

(1) The total energy absorbed by the member during the entire compression process W;

(2) The average axial load of the component during the entire compression process, $P_{\mathrm{ave}}$:(3)$$P_{\mathrm{ave}}=\frac{W}{\delta }=\frac{\int_{0}^{\delta }Pd\delta }{\delta }$$
where *P* is the load value at a certain moment and *δ* is the corresponding displacement value;

(3) The compression force efficiency is *C*_EF_, which is the ratio between the average load $P_{\mathrm{ave}}$ and the first peak load $P_{max}$:(4)$$C_{\mathrm{EF}}=\frac{P_{\mathrm{ave}}}{P_{max}}$$
the higher the compression force efficiency, the better it describes the energy absorption performance of the component. The energy absorption-displacement curves of members of different lengths are shown in Figure 14. It can be determined that with the increase in loading displacement, the energy absorbed by an empty steel tube and foam-filled steel tube members has been increasing, and the curves show an approximately linear trend. However, the energy-displacement curves of the empty steel tube members are more coincident, which indicates that the change in the length of the members will not affect the total energy absorbed by the members in the process of axial compression. The similarity of the energy absorption-displacement curve of foam-filled steel tubes decreased, and the total energy absorbed showed a trend of a slight increase. The total energy absorbed by packed and unpacked steel tube members of different lengths is shown in Figure 15. After the steel tube is filled with foam metal, it can be concluded that the total energy absorbed by the member will be greatly improved.

The energy absorption properties of different component lengths are shown in Table 5. For the component lengths of 100, 150, 200, and 250 mm, the energy absorption increases of the member after the foam-fill are 51.9%, 55.7%, 57.6%, and 57.9%, respectively. Through the comparison of the average load, it can be seen that the average load of the foam-filled steel tube was significantly increased compared with that of the empty steel tube, with a growth rate of 51.4%, 55.1%, 53.4%, and 59.1%, respectively. This indicates that the bearing capacity of the foam-filled steel tube after deformation has been significantly improved, which is also consistent with the mechanical characteristics of the foam-filled metal in its three stages, namely the elastic stage, the stress platform stage, and the compaction stage. The compression force efficiency of the foam-filled metal tube with the same length is greatly improved compared with that of an empty steel pipe: 48.9%, 52.1%, 56.7%, and 54.1%, which indicates that the filling of foam-filled metal greatly enhances the energy absorption efficiency of the steel tube.

### 4.2. Comparative Analysis of the Wall Thickness of Different Components

#### 4.2.1. Comparative Analysis of Carrying Capacity

The section and length of the simulated member are 40 mm × 40 mm and 150 mm, respectively, and the density of foam-filled metal is 270 kg/m^3^. The wall thicknesses of the simulated galvanized steel pipe members are 1 mm, 1.25 mm, 1.5 mm, 1.75 mm, and 2 mm, respectively. To simulate the test state of quasi-static loading, a 50 mm displacement load was applied to the top pressing component, and the loading speed was kept at 1 mm/s.

The comparison of the simulated axial force-displacement curves is presented in Figure 16. As can be seen from Figure 16a, the peak load of the member promotes significantly with the increase in wall thickness, and the increase in wall thickness has the most prominent contribution to the increase in carrying capacity. Combining with Figure 16b, the value of the deformed foam-filled steel tube is close to that of the first peak load. This indicates that the post-buckling carrying capacity of the foam-filled steel tube does not decrease rapidly all the time with the increasing wall thickness. On the contrary, the post-buckling carrying capacity of the steel tube will be greatly enhanced by the filling of the foam-filled metal. 

The variation trend of the peak load of different members with different wall thicknesses is shown in Figure 17, where ET represents the peak load of the empty steel tube, FFT represents the peak load of the foam-filled tube, and 1, 2, 3, and 4 correspond to the first, second, third, and fourth peaks of load, respectively. It can be seen from the figure that the peak load approximately presents a linear growth relationship with the increase in wall thickness, which indicates that the wall thickness of the member has a positive influence on the carrying capacity. Comparing the first and second peak loads, it is concluded that there is little difference between the two types of members. However, the third peak load and the fourth peak load are significant differences from the increase in load, which confirms that the bearing capacity of the foam-filled metal component is improved after buckling and deformation.

#### 4.2.2. Comparative Analysis of Carrying Capacity

The deformation patterns of 1.5 mm wall thickness empty steel tubes and aluminum foam-filled steel tubes are compared. The deformation pattern is shown in Figure 18. The combination of Figure 18a,b shows that the deformation of the empty steel tube is staggered when folded, and there are obvious cavities in the deformation. The deformation develops from the dense form to the non-dense form. However, the foam-filled steel tube will always maintain a dense and symmetrical deformation form to ensure its stability under axial stress.

The stress nephogram of the component with different wall thicknesses is shown in Figure 19, from which it can be understood that stress reflects highly in the part with the greatest deformation, and the maximum stress occurs at the corner of the section. This indicates that the stress concentration phenomenon will occur during the angle transition and that the corresponding failure is more prone to occur. It can be concluded from the diagram that the large stress region of the foam-filled steel tube is smaller than that of the empty steel pipe, and the stress concentration of the aluminum foam is also mainly concentrated on the end of deformation.

#### 4.2.3. Comparative Analysis of Energy Absorption Capacity

The energy absorption-displacement curves of components with different wall thicknesses are shown in Figure 20. It can be seen from the figure that the energy absorption-displacement curves of members with different wall thicknesses are relatively dispersed, but the curves all show a linear growth trend. Compared with the empty steel tube, the energy absorption of the aluminum foam-filled steel tube increases obviously under the same wall thickness. Combined with Figure 21, this shows that with the increase in wall thickness, the total energy finally absorbed by the empty steel tube components and the foam-filled steel tube components also reflects a linear growth trend. It is significant that the total energy absorption of the component is enhanced when it is filled with metal foam.

## 5. Conclusions

In this paper, through experimental research for the aluminum foam-filled steel tube structure, the influence of specimen length on failure mode and bearing capacity of components is obtained. Then, LS-DYNA finite element software was used to simulate the foam-filled member numerically, and the deformation and axial force-displacement curves obtained by the test were compared to verify the reliability of the numerical simulation method. On this basis, carrying capacity is affected by length and wall thickness; deformation and energy consumption of the components are discussed through numerical simulation with the empty steel tube and foam-filled steel tube as the research objects. The main conclusions are as follows: 

(1) After the axial compression test on the aluminum foam-filled steel tube components, it was determined that the deformation did not change with the growth of the component length and the axial displacement of each fold was the same. The peak load and the average load decreased with the increase in the specimen;

(2) The deformation of the empty steel tube member appears in the form of the staggered cascade under axial compression, but the size of the fold is larger. The deformation of the foam-metal-filled member becomes compact and strengthens on a regular basis. It can be concluded that the foamed metal presents a distinct inhibiting effect on the lateral deformation of the steel tube;

(3) The large stress area of empty steel tube members is wide when axial compression is applied. The large stress area decreases after filling the foam metal, and the maximum stress value also decreases, but it exceeds the yield strength of steel. At this time, the member has entered the stage of plastic energy dissipation. The maximum stress appears at the corner of the section, which indicates that the stress concentration exists in the rectangular members. The stress of foam metal exceeds the yield limit and has entered the stage of destructive energy dissipation;

(4) The first peak load of empty steel tube members decreases slightly with the increase in length, but the total energy absorption does not change, and the average carrying capacity and compression force efficiency are improved. With the increase in length of the member, the first peak load of the member filled with foamed metal also presents a slight downward trend, but the carrying capacity of the foam-filled member is significantly higher than that of the empty steel tube at the later loading stage. The total energy absorption increases with the growth of the length, which is also higher than that of empty steel tube members. The average load and compression force efficiency are increasing and have been greatly improved compared with the empty steel tube members;

(5) The mechanical properties of the empty steel tube members, including the first peak load, total energy absorption, average load, and compressive force efficiency, increase with the growth of wall thickness. At the same time, the aluminum foam-filled steel tube members present the same trend, and their value is distinctly improved compared with the empty steel tube members.

## Figures and Tables

**Figure 1 materials-16-04485-f001:**
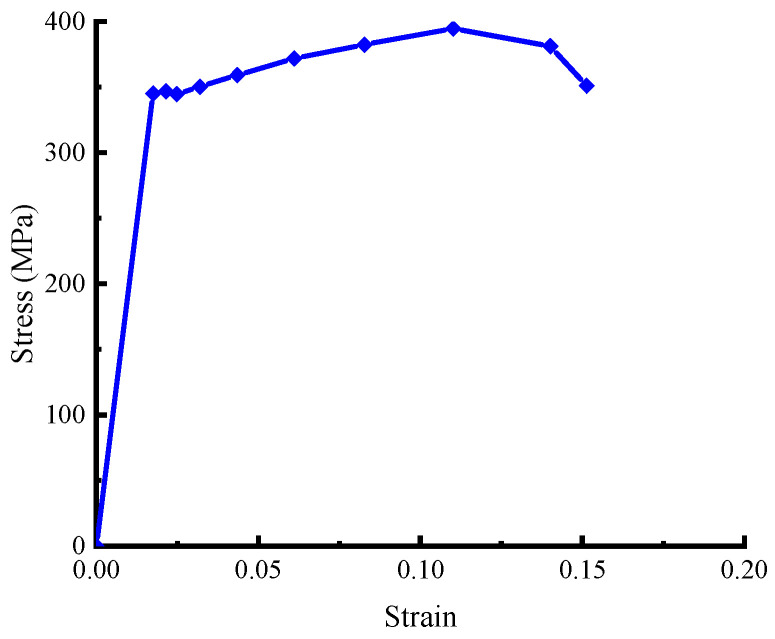
Stress-strain curve of galvanized steel.

**Figure 2 materials-16-04485-f002:**
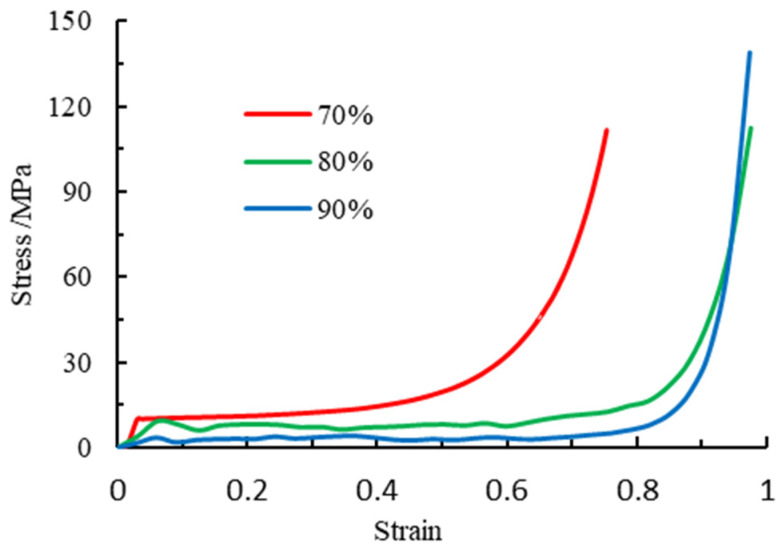
Stress-strain curves of aluminum foams with different porosities Pr.

**Figure 3 materials-16-04485-f003:**
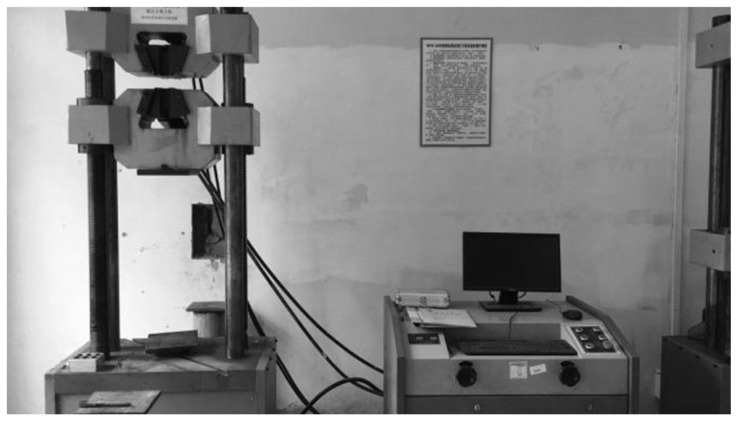
Universal testing machine.

**Figure 4 materials-16-04485-f004:**
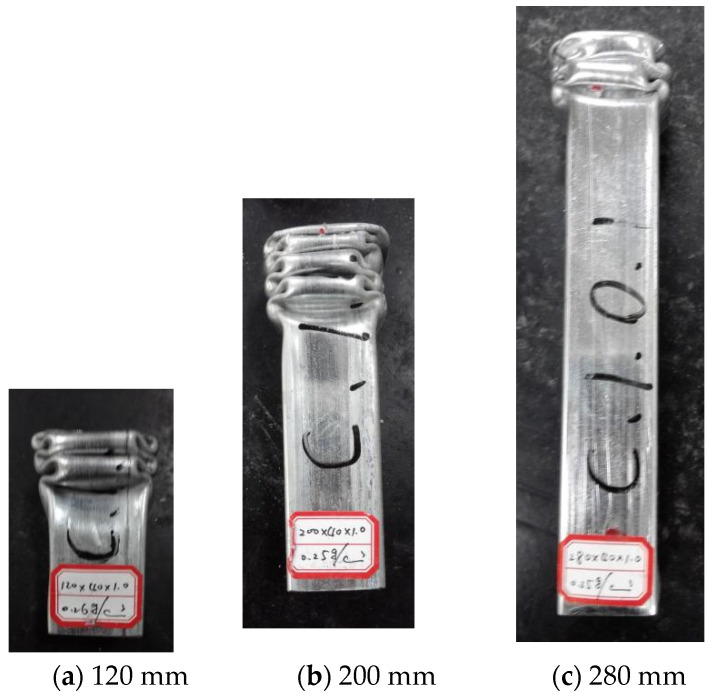
(**a**–**c**) Deformation figures of members with different lengths.

**Figure 5 materials-16-04485-f005:**
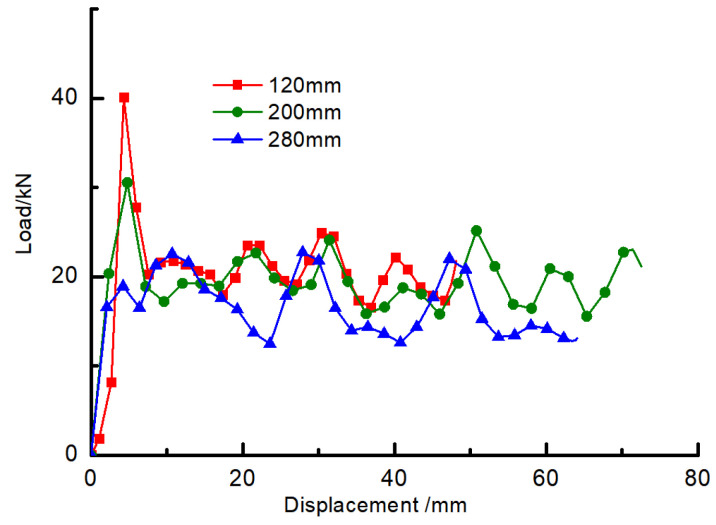
Axial force-displacement curves of specimens with different lengths.

**Figure 6 materials-16-04485-f006:**
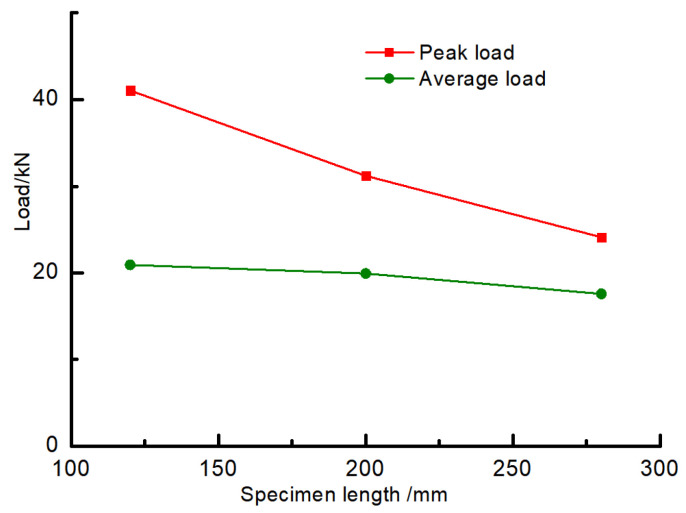
Comparison of peak load and average load of different length specimens.

**Figure 7 materials-16-04485-f007:**
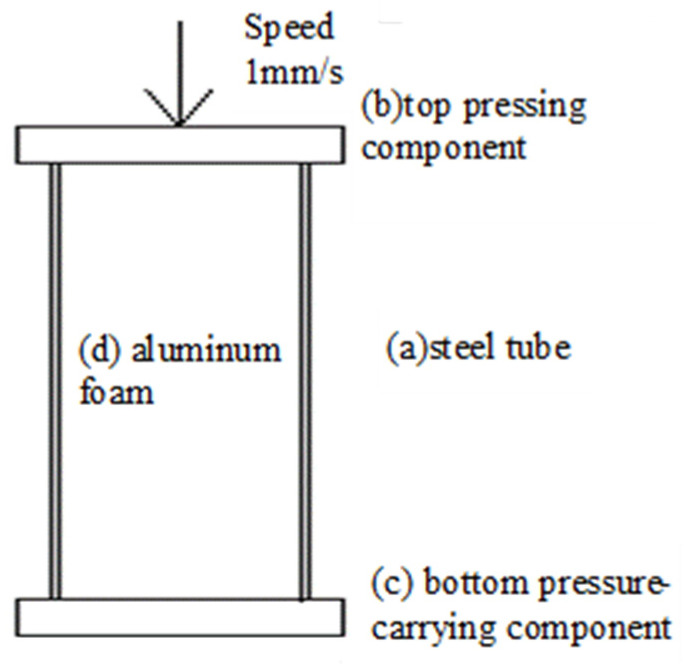
Axial compression model of foam-filled steel pipe.

**Figure 8 materials-16-04485-f008:**
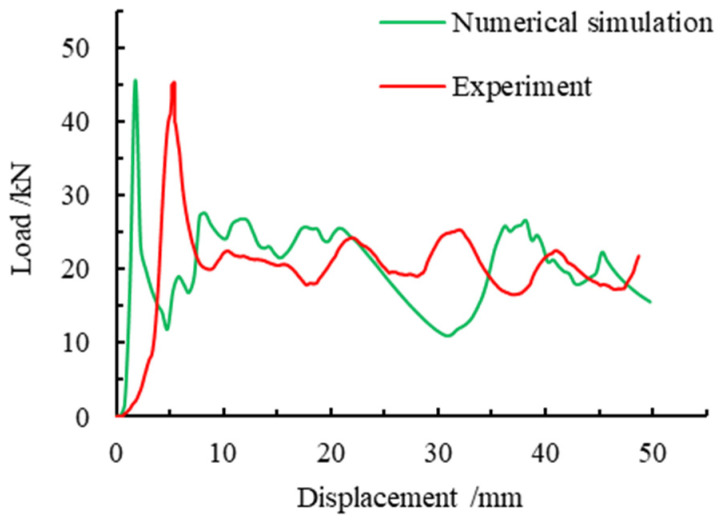
Comparison of axial force-displacement curves between numerical simulation and experimental results.

**Figure 9 materials-16-04485-f009:**
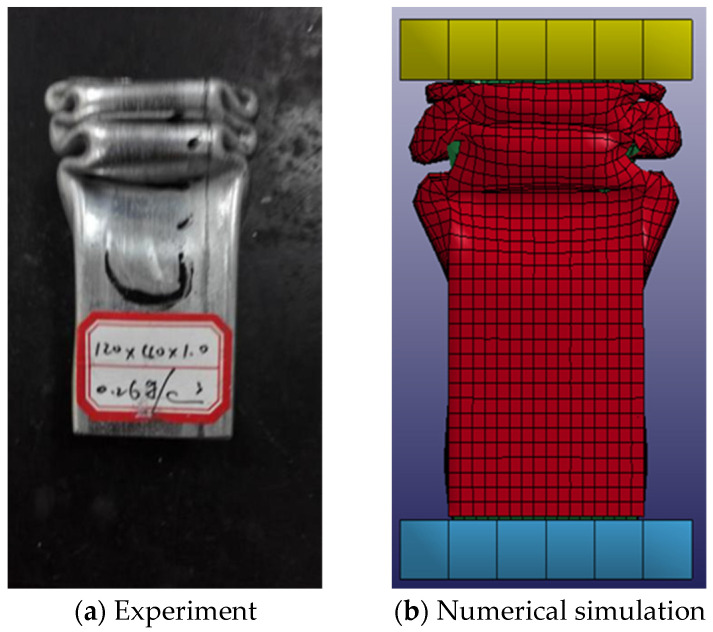
Comparison of numerical simulation and experiment deformation.

**Figure 10 materials-16-04485-f010:**
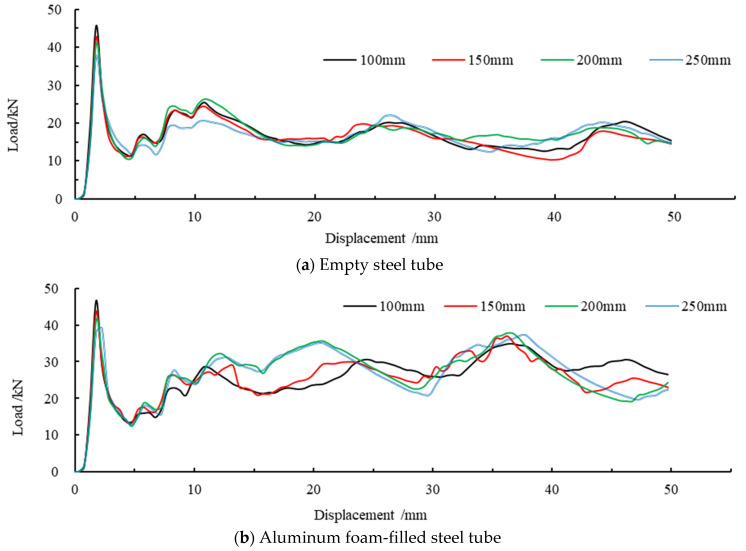
(**a**,**b**) Comparison of force-displacement curves of different member lengths.

**Figure 11 materials-16-04485-f011:**
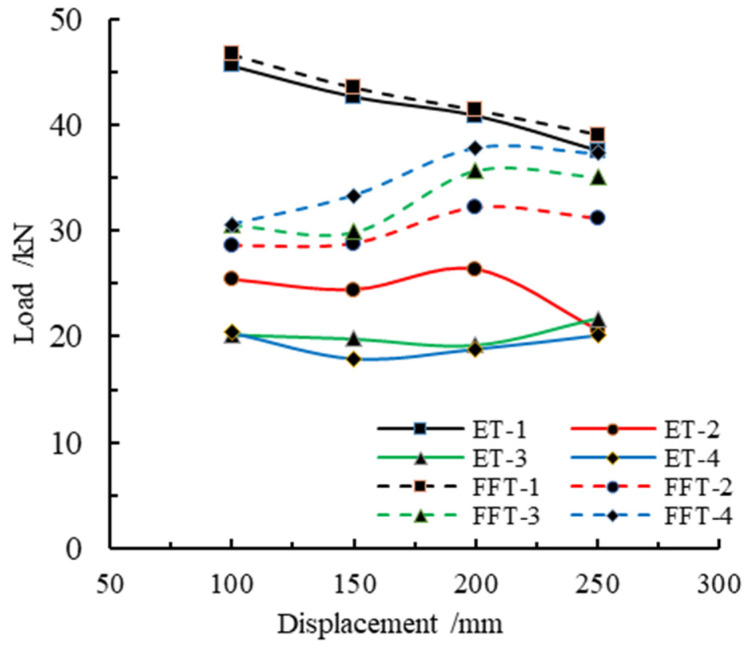
Peak load trend variation at various member lengths.

**Figure 12 materials-16-04485-f012:**
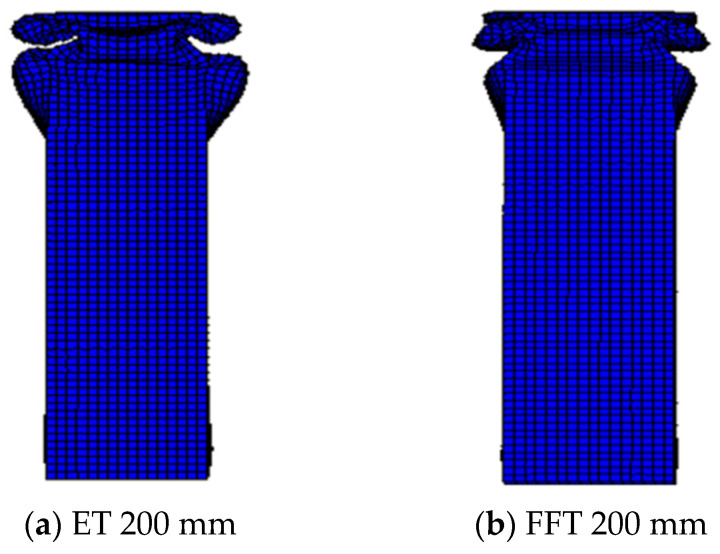
(**a**,**b**) Deformation figure for 200 mm length members.

**Figure 13 materials-16-04485-f013:**
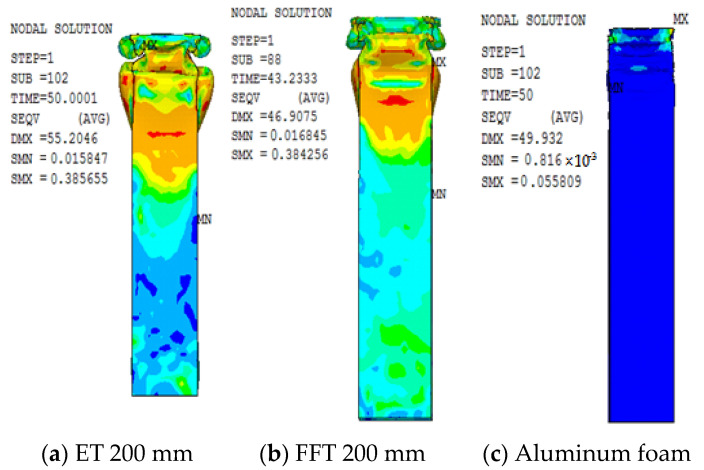
(**a**–**c**) Stress cloud figure for 200 mm length members (unit: GPa).

**Figure 14 materials-16-04485-f014:**
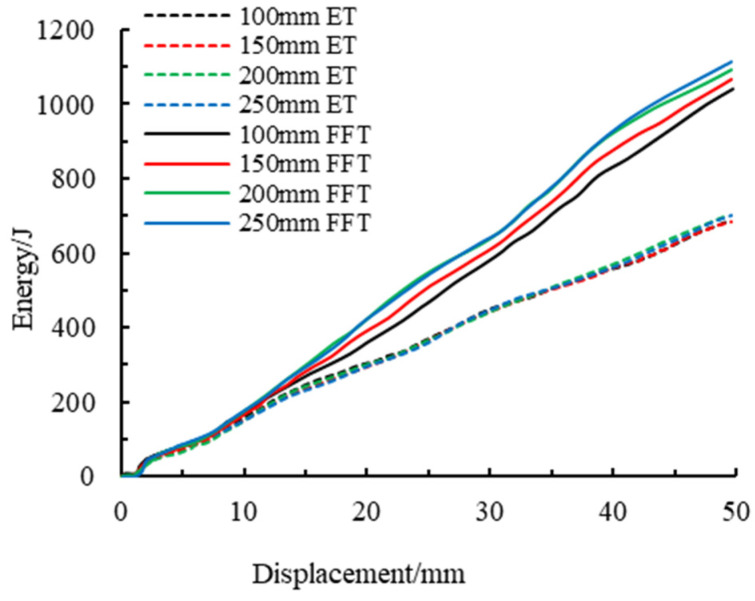
Energy absorption-displacement curves at different lengths of components.

**Figure 15 materials-16-04485-f015:**
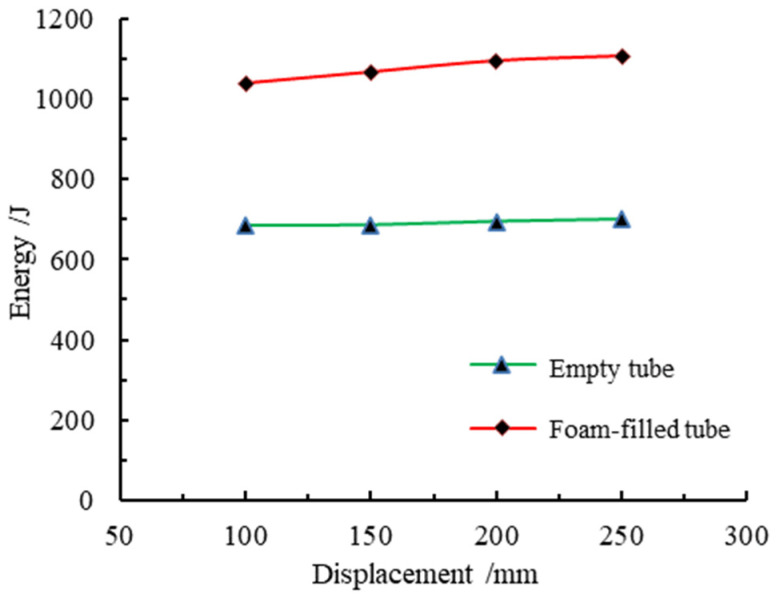
Total energy absorbed by different lengths of components.

**Figure 16 materials-16-04485-f016:**
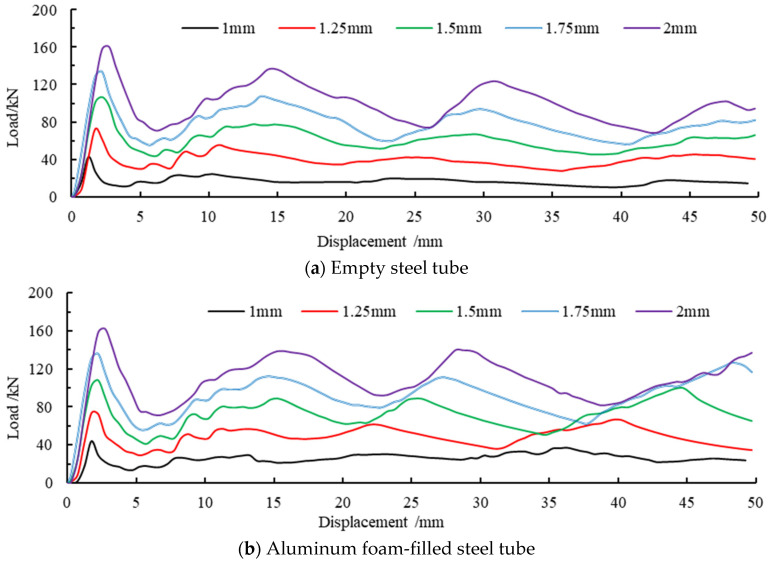
(**a**,**b**) Comparison of force-displacement curves with different wall thickness member.

**Figure 17 materials-16-04485-f017:**
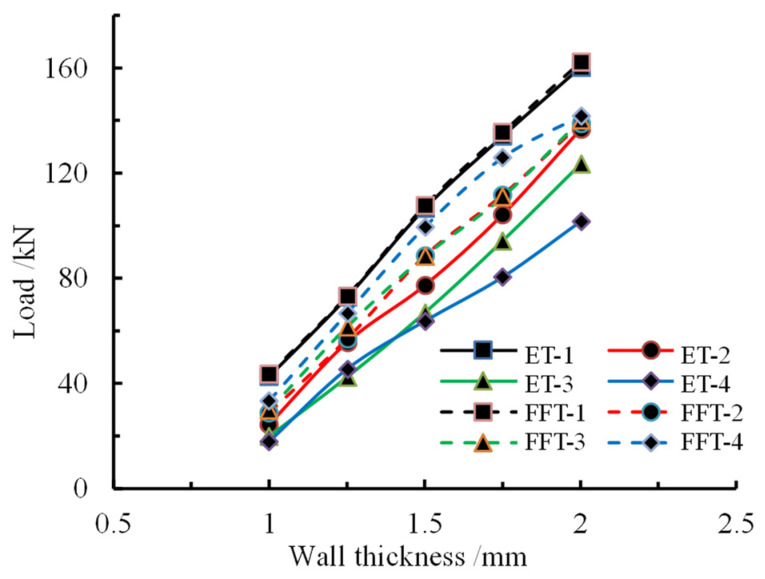
Variation in trend of peak load for different wall thicknesses.

**Figure 18 materials-16-04485-f018:**
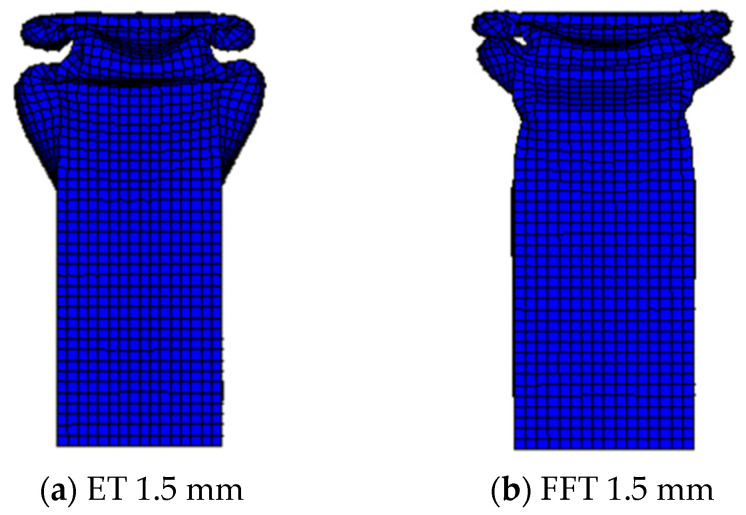
(**a**,**b**) Deformation figures for 1.5 mm wall thickness members.

**Figure 19 materials-16-04485-f019:**
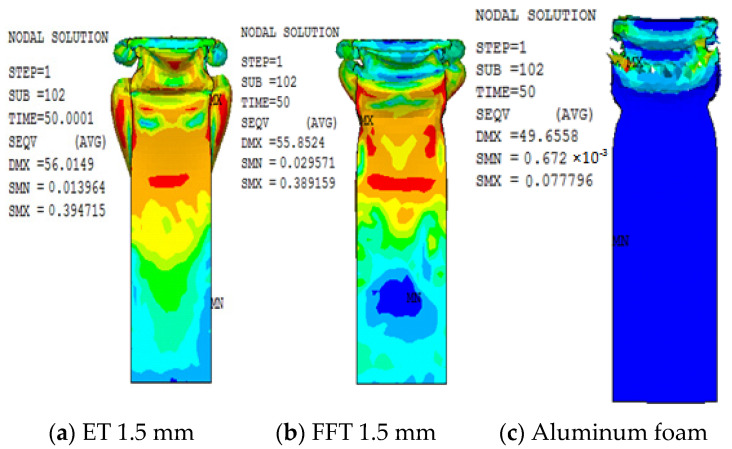
(**a**–**c**) Stress cloud figures of 1.5 mm wall thickness members (unit: GPa).

**Figure 20 materials-16-04485-f020:**
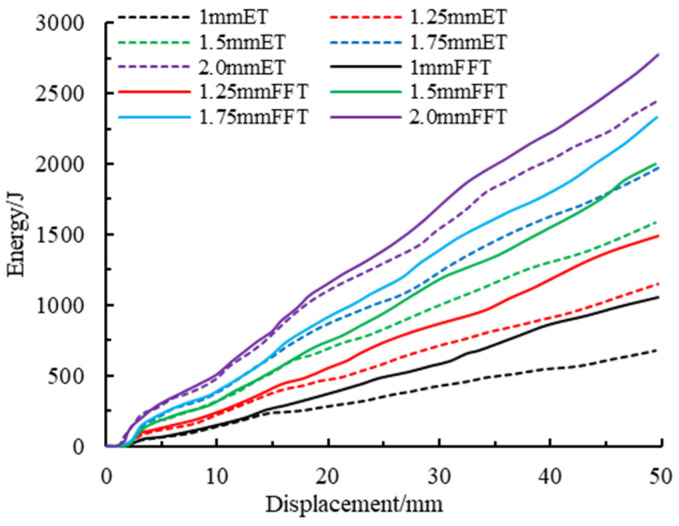
Energy absorption-displacement curves at different wall thicknesses of members.

**Figure 21 materials-16-04485-f021:**
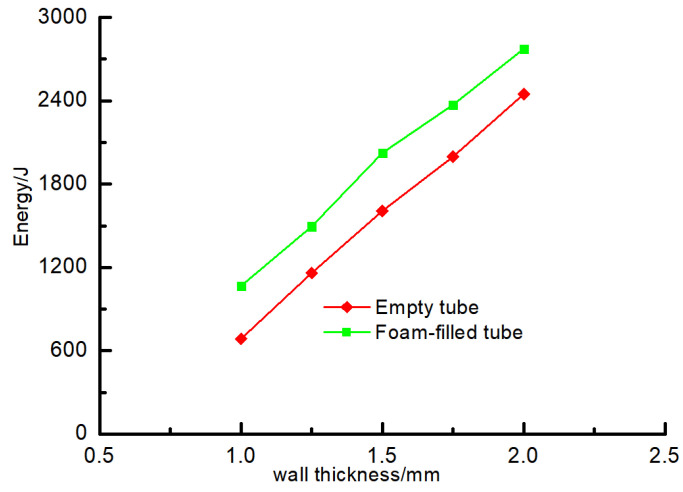
Total energy absorbed by members with different wall thicknesses.

**Table 1 materials-16-04485-t001:** Material parameters of galvanized steel tube.

Specimen	Galvanized Steel Tube
Density *ρ* (kg/m^3^)	7850
Elastic modulus *E* (GPa)	206
Poisson’s ratio *μ*	0.28
Yield stress *σ*y (MPa)	345
Ultimate stress *σ*u (MPa)	470

**Table 2 materials-16-04485-t002:** Mechanical parameters of aluminum foams with three different porosity ratios.

Porosity Ratios *P_r_*	Elasticity Modulus/MPa	Yield Strength/MPa
70%	530	12.73
80%	157	9.42
90%	76	3.8

**Table 3 materials-16-04485-t003:** Material mechanics parameters of a foam-filled steel tube.

Name	*ρ*/kg·m^−3^	*E*/GPa	*μ*	*f_y_*/MPa	*f_u_*/MPa
a	7850	206	0.28	345	470
b	7850	206	0.28	345	470
c	7850	206	0.28	345	470
d	270	0.076	0.01	3.8	64

Note: *ρ* is for density; *E* is for elasticity modulus; *μ* is for Poisson’s ratio; $f_{y}$ is for yield strength; and $f_{u}$ is for ultimate strength.

**Table 4 materials-16-04485-t004:** Peak load at different lengths of components.

Material	Peak Load	Length/mm			
		100	150	200	250
Empty steel tube/kN	ET-1	45.64	42.71	40.93	37.62
	ET-2	25.44	24.46	26.39	20.65
	ET-3	20.17	19.79	19.2	21.73
	ET-4	20.41	17.86	18.8	20.15
Aluminum foam-filled steel tube/kN	FFT-1	46.72	43.56	41.41	39.07
	FFT-2	28.58	28.78	32.24	31.21
	FFT-3	30.62	29.92	35.72	35.13
	FFT-4	30.63	33.38	37.89	37.35
Peak difference value/%	Peak-1	2.4	2.0	1.2	3.9
	Peak-2	12.3	17.7	22.2	51.1
	Peak-3	51.8	51.2	86.1	61.7
	Peak-4	50.1	86.9	101.5	85.4

**Table 5 materials-16-04485-t005:** Evaluation of energy absorption performance at different length components.

Component	Length/mm	Total Energy/J	Average Load/kN	Compression Force Efficiency/%
Empty steel tube/kN	100	683.66	17.11	37.2
	150	685.17	16.41	38.4
	200	694.85	17.36	42.4
	250	700.85	16.73	44.4
Aluminum foam-filled steel tube/kN	100	1038.76	25.91	55.4
	150	1066.83	25.45	58.4
	200	1094.91	26.63	64.3
	250	1106.58	26.73	68.4
Value difference/%	100	51.9	51.4	48.9
	150	55.7	55.1	52.1
	200	57.6	53.4	51.7
	250	57.9	59.8	54.1

## Data Availability

Not applicable.

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
