# Peer review of "Study on Quasi-Static Axial Compression Performance and Energy Absorption of Aluminum Foam-Filled Steel Tubes"

_materials, 2023, doi:10.3390/ma16124485_

Round 1
Reviewer 1 Report
The manuscripts illustrates an experimental and numerical study on the response of aluminium foam-filled steel tubes subjected to axial compression tests with special attention to energy absorption characteristics. First, a small set of specimens were tested in the laboratory and finite element analyses were conducted to simulate the tests. Given the good matching between experiments and simulations, the latter were used to conduct a wider parametric study.
The study appears to have been correctly designed and conducted. Results are worth of publication. However, English language needs a general revision.
Also, some minor changes are suggested:
- it could be worth emphasising in the title that focus is on energy absorption characteristics, e.g. adding "for energy absorption" at the end of current title;
- lines 212 and 213: the titles of Section 4.1 and Subsection 4.1.1 are the same; please, correct;
- line 296 and following: symbol E for energy has already been used for the Young's modulus; please, use different symbols.
Author Response
Point 1: It could be worth emphasising in the title that focus is on energy absorption characteristics, e.g. adding "for energy absorption" at the end of current title.
Response 1: Change it to “Study on quasi-static axial compression performance and en-ergy absorption of aluminum foam-filled steel tubes”.
Point 2: Lines 212 and 213: the titles of Section 4.1 and Subsection 4.1.1 are the same; please, correct.
Response 2: The titles of Subsection 4.1.1 is changed to “Comparative analysis of bearing capacity”.
Point 3: Line 296 and following: symbol E for energy has already been used for the Young's modulus; please, use different symbols.
Response3: Change E to W

Reviewer 2 Report
The paper describes an experimental and numerical investigation into the ultimate strength behavior of aluminium foam-filled steel tubes. The paper is relatively long regarding the scope of results. It requires major modifications according to the comments below.
1. The reviewer does not understand why the foam porosity has not been varied. This has much more influence than e.g. the tube length!
2. When citing, give only family names!
3. Line 54: 'multi-layer aluminium tubes' cannot immediately be understood. Give some explanation.
4. Caption of Fig. 1: Add 'Pr' at end (link to equation above). Also in the left of Table 1!
5. Section 1: Any data of the steel material is missing! This includes also the nonlinear material law which is used later.
6. Also a photo or sketch of the test set-up is missing!
7. Caption of Fig. 9: It is helpful to mention 'ET = empty and EFT = filled tubes'. Also in Fig. 15.
8. Line 254ff: Two digits behind dot is too much for percentages.
9. Table 3: 1, 2, ... should read 1st peak, 2nd peak ... (also Table 4).
10. Line 284: The reviewer does not understand the sentence (what is meant with 'angle transition'?).
11. E is used for two variables in the paper, which has to be avoided.
The English wording and grammar are poor. The help of an interpreter or native speakter is highly recommended. Examples:
- 'alumunium foam-filled steel tube and empty steel tube' is repeated very often. It can be shortened, e.g. by 'such tubes' in line 10.
- Delete all 'Meanwhile', because this word is used in wrong sense. 'Meanwhile' is connected with 'time'.
- Line 39: compression => compressive.
- et al => et al. (several times).
- Line 81: analyzed => analyzes.
- Line 88: Sentence misses main part! Also sentence in line 317.
- Line 107: Add 'rectangular' before 'section'!
- Line 122: by axial => by the axial (tests were already mentioned!).
- Line 123: load => loads; line 124: decreases => decrease.
- line 150: has meshed => has been meshed.
- Line 158: Several spaces are missing!
- Line 212/213: Headings should not have the same name!
- Line 226: present => presents.
- Table 3: growth range => growth (why range?)
- Line 269: present => is (or presents).
- Line 301: The structure of the sentence is unclear. Is a colon missing?
- Line 306: Delete 'change' (also later).
- Line 309: The curve ... is separated? How can this happen?
- Line 331: differ-rent => different.
- Line 355: is => show.
- Line 364: form of staggered => staggered form?
- Line 368: stress cloud ... of the wall thickness???
- Line 369: reflect => reflects.
- Line 383: reflects => , this shows.
- Line 385: It's => It is.
- Fig. 19: tub => tube.
- Line 392: experiment => experimental.
- Line 393: verb is missing.
- Line 404: appear => appears.
Author Response
Point 1: The reviewer does not understand why the foam porosity has not been varied. This has much more influence than e.g. the tube length!
Response 1: Due to the limitation of space, the effect of the bubble will be discussed in another paper.
Point 2: When citing, give only family names!
Response 2: Modified
Point 3: Line 54: 'Multi-layer aluminium tubes' cannot immediately be understood. Give some explanation.
Response 3: A smaller aluminum tube is inserted inside the aluminum tube.
Point 4: Caption of Fig. 1: Add 'Pr' at end (link to equation above). Also in the left of Table 1!
Response 4: Modified
Point 5: Section 1: Any data of the steel material is missing! This includes also the nonlinear material law which is used later.
Response 5: Added
Point 6: Also a photo or sketch of the test set-up is missing!
Response 6: Added
Point 7: Caption of Fig. 9: It is helpful to mention 'ET = empty and EFT = filled tubes'. Also in Fig. 15.
Response 7: Modified
Point 8: Line 254ff: Two digits behind dot is too much for percentages.
Response 8: Modified
Point 9: Table 3: 1, 2, ... should read 1st peak, 2nd peak ... (also Table 4).
Response 9: Modified
Point 10: Line 284: The reviewer does not understand the sentence (what is meant with 'angle transition'?).
Response 10: This indicates that the stress concentration phenomenon will appear at the four right angles of the rectangular section member.
Point 11: E is used for two variables in the paper, which has to be avoided.
Response 11:Change E to W
Point 12: Line 39: compression => compressive.
Response 12: Modified
Point 13: et al => et al. (several times).
Response 13: Modified
Point 14: Line 81: analyzed => analyzes
Response 14: Modified
Point 15: Line 88: Sentence misses main part! Also sentence in line 317.
Response 15: Modified
Point 16: Line 107: Add 'rectangular' before 'section'!
Response 16: Modified
Point 17: Line 122: by axial => by the axial (tests were already mentioned!).
Response 17: Modified
Point 18: Line 123: load => loads; line 124: decreases => decrease.
Response 18: Modified
Point 19: Line 150: has meshed => has been meshed.
Response 19: Modified
Point 20: Line 158: Several spaces are missing!
Response 20: Modified
Point 21: Line 212/213: Headings should not have the same name!
Response 21: Modified
Point 22: Line 226: present => presents.
Response 22: Modified
Point 23: Table 3: growth range => growth (why range?)
Response 23: Modified
Point 24: Line 269: present => is (or presents).
Response 24: Modified
Point 25: Line 301: The structure of the sentence is unclear. Is a colon missing?
Response 25: Modified
Point 26: Line 306: Delete 'change' (also later).
Response 26: Modified
Point 27: Line 309: The curve ... is separated? How can this happen?
Response 27: Modified
Point 28: Line 331: differ-rent => different.
Response 28: Modified
Point 29: Line 355: is => show.
Response 29: Modified
Point 30: Line 364: form of staggered => staggered form?
Response 30: Modified
Point 31: Line 368: stress cloud ... of the wall thickness???
Response 31: Modified
Point 32: Line 369: reflect => reflects.
Response 32: Modified
Point 33: Line 383: reflects => , this shows.
Response 33: Modified
Point 34: Line 385: It's => It is.
Response 34: Modified
Point 35: Fig. 19: tub => tube.
Response 35: Modified
Point 36: Line 392: experiment => experimental.
Response 36: Modified
Point 37: Line 393: verb is missing.
Response 37: Modified
Point 38: Line 404: appear => appears.
Response 38: Modified

Reviewer 3 Report
Porous and foam materials are often used as structural elements to increase energy absorption capacity. In this regard, the work is devoted to an urgent topic.
Several typos were found in the article:
1. Line 146 «Cowper-Symbols» should be Cowper-Symonds [Cowper G.R., Symonds P.S. Strain hardening and strain rate effects in the impact loading of cantilever beams. Brown University Appl. Math. Report, 28 (1958) pp. 1-46.]
2. Line 331 «differ-rent» should be different
I want to make a number of recommendations that, in my opinion, will improve the quality of the article:
The legend of figures 11, 17 is poorly readable. There is no decryption of the data given in the legend. It is necessary either to remove the excess from the legend, or to decipher the meaning in the text. The text also does not decipher which stress fields are shown in the figures.
The model of compressible foam is not sufficiently described. The elastic part is given. There is no description of the «volumetric strain-pressure» equation, which is a very important part of the equation of state for compressible materials.
Table 2 shows the general characteristics of the material. It is not clear how these data are embedded in the previously described models in the form of parameters.
In my opinion, Table 2 can be shortened, since positions a, b, c are the same material. It would be more logical to give the characteristics of materials in the table: 1 – steel, 2 – porous aluminum. In the text, indicate that the parts of the model a, b, c are made of material 1, part d is made of material 2.
It is not clear why an explicit LS-DYNA solver scheme is used to model a quasi-static process. In LS-DYNA solver, an implicit scheme for integrating equations in time is implemented for such problems.
The structure of the article seems a little out of wholeness. The first part presents the results of experimental studies, which are not used in any way in the future. It would be interesting to see a comparison of the simulation results with the results of the authors' experimental study. In particular, to compare the experimental dependences of Figure 4 with the results of numerical simulation.
In general, the article contains interesting information. A large number of numerical studies have been performed.
Author Response
Point 1: Line 146 «Cowper-Symbols» should be Cowper-Symonds [Cowper G.R., Symonds P.S. Strain hardening and strain rate effects in the impact loading of cantilever beams. Brown University Appl. Math. Report, 28 (1958) pp. 1-46.]
Response 1: Modified
Point 2: Line 331 «differ-rent» should be different
Response 2: Modified

Round 2
Reviewer 2 Report
Following comments regardin the revised paper need to be considered:
Line 64: which => where.
Line 126: Figure => Figure 1.
Check formatting of heading 2.2.
The background of the question regarding test setup was to understand the end constraints and why the buckling occurs only at the upper end. Fig. 3 is not really informative.
Author Response
Following comments regardin the revised paper need to be considered:
Line 64: which => where.
Response 1: Modified
Line 126: Figure => Figure 1.
Response 2: Modified
Check formatting of heading 2.2.
Response 3: Checked
The background of the question regarding test setup was to understand the end constraints and why the buckling occurs only at the upper end. Fig. 3 is not really informative.
Response 4: Modified
More detailed pictures are provided
